# Breeding of Highly Virulent *Beauveria bassiana* Strains for Biological Control of the Leaf-Eating Pests of *Dalbergia odorifera*

Xianpeng Ni [1], Hongjun Li [1], Yandong Xia [1], Yan Lin [2], Chuanting Wang [2], Cong Li [1], Junang Liu [1,*] and Guoying Zhou [1]

1 Key Laboratory of National Forestry and Grassland Administration for Control of Diseases and Pests of South Plantation, Central South University of Forestry and Technology, Changsha 410004, China
2 Chengmai State-Owned Forest Farm, Chengmai County 571939, China
* Correspondence: kjc9620@163.com

**Abstract:** *Dalbergia odorifera* (*D. odorifera*), commonly named the fragrant rosewood, is one of the second-level protected wild plants in China, and one of 34 species of rosewood in five genera and eight categories in the National Standard of China. As a kind of traditional Chinese medicine (TCM), it plays an important role in the pharmaceutical industry, including the treatment of cardiovascular diseases, rheumatic pain, etc. With the continuous expansion of the planting area of *D. odorifera*, the diseases and pests of *D. odorifera* become more and more serious, among which leaf-eating pests are the most serious. In this study, ultraviolet rays and microwaves were used to mutagenize *Beauveria bassiana* (*B. bassiana*) strain HNCMBJ-P-01, and excellent mutant strains with high spore yield and high virulence were screened out, and then they were prepared into a wettable powder for forest control experiments to study their biocontrol effects. The virulence screening test showed that the virulence of strain HBWB-44 was the strongest, and the 10 day corrected mortality rate was 80.00%, and the lethal time was 5.622 days. The results of biological control test showed that the control effect of *B. bassiana* wettable powder 100 times solution reached 60.89%, second only to the botanical fungicide matrine. Generically, The *B. bassiana* that we screened and mutated showed a good killing effect on *Plecoptera bilinealis* (*P. bilinealis*), and the wettable powder produced by it showed a good control effect on the leaf-eating pests of *D. odorifera*. The application of fungal insecticides in plantations has a good prospect for controlling the occurrence of leaf-eating pests of *D. odorifera*.

**Keywords:** fungi; insects; biological control; *B. bassiana*



## 1. Introduction

*Dalbergia odorifera* (*D. odorifera*), or fragrant rosewood, *Huanghuali*, etc., belongs to Leguminosae, Papilionoideae, Dalbergia, semi-deciduous trees [1], and is one of the second-level protected wild plants in China, and the redwood species consists of 33 defined tree species belonging to five genera and eight categories by the Chinese Standard [2]. Precious rosewood species *D. odorifera* is native to Hainan and is mainly distributed in the tropical areas with relatively narrow altitudes on Hainan Island [3]. It was introduced to subtropical regions such as Guangdong, Fujian, and Guangxi in the 1950s. It has now been introduced in Guangdong, Fujian, Guangxi, and other provinces [4]. As a kind of traditional Chinese medicine, it is also known as "Ginger Fragrance" and contains a series of chemical components such as flavonoids [5], phenols [6], and sesquiterpenes derivatives [7], which play an important role in the pharmaceutical industry for the treatment of cardiovascular diseases, cancer, diabetes, blood disorders, ischemia, swelling, and rheumatic pain. Therefore, it has high medicinal and commercial value [8].

At present, there are relatively few research reports on the prevention and control of *D. odorifera*. With the continuous expansion of the planting area of *D. odorifera*, the damage

of leaf-eating pests is getting more and more serious [9]. Xiang Tao et al. carried out a comprehensive investigation on the main pests of a *D. odorifera* plantation in Chengmai State-owned Forest Farm in Hainan, China. The results showed that 66% of the trees were damaged by the pest insects, with the most serious damage to leaves [9]. Zhang Wei et al. pointed out in the pest investigation report of *D. odorifera* planting areas in different counties of Hainan Province that there were nearly 19 main pests of *D. odorifera* in these areas, e.g., *P. bilinealis*, *Anomala cupripes* Hope, and *Lawana imitata* Melichar, which were found in different parts of the trees, with the largest number of pests distributed in the leaves [10]. Previous studies on the morphological structure, biological characteristics, and epidemic regularity of different leaf-eating pests, such as *P. bilinealis*, *Plecoptera subpallida*, *Plecoptera oculata* Moore, etc. [11–13], provided a basis for controlling pest damage of *D. odorifera*.

Microbial pesticides can effectively replace chemical pesticides. Microbial toxins can be defined as biotoxic substances derived from microorganisms, such as bacteria, nematodes, fungi or protozoa. The pathogenic effects of these microorganisms on target pests are species-specific. Fungal pesticides play an important role in the biological control of pathogens and insect pests in agriculture and horticulture [14]. *B. bassiana* plays a key role in the control of many agricultural and forestry pests. At the beginning of the 20th century, the first biological agent with *B. bassiana* as the main active ingredient was successfully registered in the United States [15]. By the end of 2007, the insecticides with *B. bassiana* accounted for one third of the global fungal insecticides [16]. Zhang used *B. bassiana* granules and *Stratiolaelaps scimitus* to successfully improve the control effect on *Frankliniella occidentalis* [17]. Deborah et al. mixed *B. bassiana* spores with vegetable oil to make the death rate of pollen beetles much higher than expected [18]. The combination of *B. bassiana* spores and *Trichoderma lignorum* as a bioinsecticide controlled *Atta cephalotes* [19]. At present, *B. bassiana* has been widely used in the control of insects such as the corn borer, *Monochamus alternatus*, and silkworm larvae [20–22].

Microbial mutation breeding is the artificial induction of microbial gene mutation. It is a change in the genetic structure or function of microorganisms. After conditional screening, specific mutants can be obtained. Strains isolated from nature often cannot meet the requirements of industrial production [23,24]. Ultraviolet light is a common mutagenesis method. Ultraviolet light has a strong genotoxic effect, which can cause DNA damage and induce mutation. Ultraviolet mutagenesis has the characteristics of low cost, simple operation, high mutation rate, and high safety. It is a common mutagenesis method that can obtain a large number of mutations in a short time [25–27]. Microwave is regarded as kind of physical mutagen. Pinakin et al. carried out microwave mutagenesis on *Bacillus brevis* and obtained four mutants with a high yield of cellulase enzymes [28]. Feng Jie screened the mutant strain CB-27 by ultraviolet and microwave mutagenesis from *B. bassiana* Bb111. The corrected death rate of CB-27 on the camellia weevil reached 86.83% [29].

At present, chemical control is the main method to control the pests of *D. odorifera*. Long-term use of chemical pesticides not only leads to drug resistance of pests, but also damages the environment. Bioinsecticides have attracted more and more attention in the management of pests. *B. bassiana* is a broad-spectrum insect pathogenic fungus, which has many successful examples in controlling agricultural and forestry pests. However, the application of *B. bassiana* in the pest control of *D. odorifera* is still relatively limited. In this study, through UV and microwave mutation breeding, we screened and obtained the mutant HBWB-44 for the first time, which demonstrated high virulence to *P. bilinealis*, the common leaf-eating pests of *D. odorifera* in Hainan, China. Especially, the forest control efficiency of the HBWB-44 wettable powder was evaluated, to investigate the potential of the strain HBWB-44 to be developed into a biological control agent (BCA) in the management of *D. odorifera* leaf-eating pests.

## 2. Materials and Methods

### 2.1. Sample Collection and Processing

The 2nd–4th instar larvae of *P. bilinealis* (collected in Chengmai State-owned Forest Farm, Chengmai County, China) were raised in a greenhouse (temperature 28 °C, humidity 80%, light–dark ratio 12:12). The feeding leaves were mainly surface-sterilized young leaves of *D. odorifera*.

*B. bassiana* HNCMBJ-P-01 was isolated and preserved by the Forestry Pathology Laboratory of Central South University of Forestry and Technology. This strain was isolated from the cadavers of *P. bilinealis* in Chengmai Forest Farm (110°32′51″~110°34′54″ E, 21°43′42″~21°44′09″ N) in Haikou City, Hainan Province, in September 2019. *B. bassiana* CXBJ-01 was isolated from the *B. bassiana* wettable powder produced by Shanxi Lvhai Pesticide Technology Co., Ltd., Taiyuan, China. *B. bassiana* CXBJ-03 was isolated from the *B. bassiana* wettable powder produced by Guangzhou Duoyuduo Biotechnology Co., Ltd., Guangzhou, China. We inoculated the *B. bassiana* strains on PDA solid medium and cultured them under constant temperature for 15 days at 28 °C and L:D = 12:12 photocycle.

### 2.2. Experimental Design

The *B. bassiana* strain HNCMBJ-P-01 was subjected to UV and microwave mutagenesis successively, and the lethal rates and positive mutation rates were recorded under different treatment time. The mutagenic strain with the highest virulence against *P. bilinealis* was screened out, and then the stability and heat resistance determinations were carried out. In addition, *B. bassiana* wettable powder was prepared to conduct the indoor virulence assay and forest control test to evaluate the biological control ability of the strain. In the forest trial, the spraying treatment was carried out every 10 days.

### 2.3. Preparation of Spore Suspensions

The cultured *B. bassiana* strains were sampled by punching a hole from the center to 1/2 of the edge of the PDA solid plate medium using a hole punch with a diameter of 5 mm, and placed in a centrifuge tube containing 10 mL of 0.05% Tween-80 sterile water. Spore suspensions were prepared by shaking and mixing in the test tube with a scroll shaker, and the number of spores per unit area was calculated using a hemocytometer. The spores were eluted with an aqueous solution and prepared into a $1 \times 10^7$ spores/mL spore suspension, which was stored in a refrigerator at 4 °C for future use. Three holes were randomly punched in each plate, and three plates were used for each treatment.

### 2.4. UV Mutagenesis and Screening of High Virulent Strains

First, 10 mL of the *B. bassiana* HNCMBJ-P-01 spore suspension with a concentration of $1 \times 10^7$ spores/mL was pipetted onto a petri dish (diameter 9 cm) on a magnetic stirrer, and placed under a 15 W UV lamp (28 cm) preheated for 30 min, irradiated for 3, 6, 9, 12, 15, 18, and 21 min. Treatment without UV lamp irradiation was set as a blank control, and each treatment was repeated 3 times.

The strain HNCMBJ-P-01 was mutagenized under the optimal ultraviolet mutagenesis conditions, and the obtained positive mutant strain was prepared in a spore suspension of $1 \times 10^7$ spores/mL. Using the dip method, the 4-year-old healthy larvae of *P. bilinealis* were dipped in the liquid for 10 s. After inoculation, the larvae were put into a sterilized plastic box and incubated at 28 °C. The death of the larvae was observed and recorded every day, and the young leaves of *D. odorifera* were regularly placed as food. The observation was continued for 10 days. The larvae treated with distilled water containing 0.1% Tween-80 was set as the control. The most virulent mutant strains were screened out for further research. The dead insect carcasses were removed and placed in sterilized petri dishes for heat preservation and moisture retention cultivation. Mycelium growth and spore formation on the carcasses were observed to verify whether the insects were killed by *B. bassiana*. Mortality rate (%) and $LT_{50}$ were calculated. Adjusted death rate = (death rate of treatment group − death rate of control group)/(1 − death rate of control group) × 100%.

### 2.5. Microwave Mutagenesis and Screening of High Virulent Strains

The mutant strains with relatively high virulence were obtained after screening by ultraviolet mutagenesis, and the spore suspension with a concentration of $1 \times 10^7$ spores/mL was prepared after sporulation by constant temperature cultivation. It was placed in a beaker filled with ice water to reduce the thermal effect of microwaves, which will overheat the suspension and kill a large number of spores. The Galanz T770D20T-TD (700 W) (Galanz Inc., Foshan, China) was used for microwave mutagenesis. The irradiation time was 15, 30, 45, 60, 75, 90, 105, and 120 in sequence. After microwave irradiation, the spore suspension was continuously diluted by the ten-fold dilution method, and 0.1 mL was inoculated on the PDA medium and cultured inversely at 28 °C for 3 days to count the colonies and calculate the fatality rate. Individual colonies were purified and cultured to produce spores, the spore production was measured, and the positive mutation rate was calculated. The optimal time of microwave mutagenesis was determined according to the fatality rate and positive mutation rate. The virulence test of the strains was similar to Section 2.4.

### 2.6. Genetic Stability and Heat Resistance Test of Mutant Strains

The screened highly virulent mutant strains, the original strain HNCMBJ-P-01, *B. bassiana* CXBJ-01, and *B. bassiana* CXBJ-03 were prepared in a spore suspension with a concentration of $1 \times 10^7$ spores/mL. A pipette gun was used to add 2 mL spore suspension into a 5 mL centrifuge tube, and the centrifuge tube was heated in a 48 °C water bath. Next, 100 μL of the spore suspension was extracted from each centrifuge tube every 6 min and transferred to a 5 mL centrifuge tube containing 11 mL of germination solution, and then cultured in an oscillator (28 ± 1 °C, 160 rpm/min) for 24 h. The respective germination rate was measured. The control was the spore germination rate of the *B. bassiana* spore suspension without water bath stress [30]. The spore suspensions of each strain were subjected to water bath stress until the spores' germination rate reached zero, and each group was repeated 3 times.

### 2.7. Indoor Virulence Assay of B. bassiana Wettable Powder

The carrier, wetting agent, dispersing agent, and UV protectant were mixed according to previous studies with appropriate modifications [31,32]. Briefly, the *B. bassiana* wettable powder formulation contained 30% conidia, 5% polyvinyl alcohol, 5% sodium polyphosphate, and 0.5% zinc oxide, with additional diatomaceous earth up to 100%. The *B. bassiana* wettable powder was prepared into 100-fold dilution and 500-fold dilution. At the same time, an appropriate amount of the *B. bassiana* wettable powder from Shanxi Lvhai Pesticide Technology Co., Ltd. was prepared into a 1000-fold solution, and the matrine insecticide from the Institute of Plant Protection, Chinese Academy of Agricultural Sciences was prepared in a 100-fold solution. The treatment solutions prepared above were used to dip the fourth instar healthy larvae of *P. bilinealis*. After dipping each larva for 10 s, *P. bilinealis* was taken out and placed on sterilized filter paper and allowed to crawl freely for 2 min. After fully draining the surface moisture, *P. bilinealis* was put into a sterilized plastic box, then the box was placed into the greenhouse at 28 °C (humidity 8.0%, 12:12 light–dark ratio), and regularly put the young leaves of *D. odorifera* as food. Each group had 3 replicates and each box had 20 larvae. The death of larvae was observed and recorded every day for 10 days. The control group was treated with distilled water containing 0.1% Tween-80. The assay was carried out in Hunan Provincial Key Laboratory for Control of Forest Diseases and Pests, Changsha, China.

### 2.8. Forest Control Effect of B. bassiana Wettable Powder

The forest control efficiency of *B. bassiana* was tested in the plantation of *D. odorifera* at Chengmai National Forest Farm in Hainan Province. The area of each sample plot was 12 m$^2$ (3 × 4). As it described previously, the midpoint of the diagonal was taken as the central sampling point, and then four points on the diagonal with the same distance from

the central point as the other sampling points were selected [33]. The five-point sampling method was used in each plot: at each of the five points, 12 trees were selected sequentially; a total of 60 trees of *D. odorifera* were involved. The population decline rate of *P. bilinealis* larvae was observed and recorded at each sample site.

### 2.9. Statistical Analyses

The mortality rates were calculated using Microsoft Office 2017 Excel (Microsoft, Redmond, WA, USA). $LT_{50}$ analyses were performed using SPSS software (Version 19.0, IBM-SPSS, Armonk, NY, USA) [34]. Statistical significance was detected by one-way ANOVA analysis followed by Duncan's multiple range tests (DMRT). *p*-value < 0.05 was considered statistically significant.

## 3. Results

### 3.1. Determination of UV Mutagenesis Time

The mortality rate increased with the increase in irradiation time. When the UV irradiation time was 21 min, the mortality rate of the UV-mutated strain was 100%. From 0 to 9 min, the positive mutation rate of the strain increased continuously with the increase in UV irradiation time. The positive mutation rate of the strain at 12 min and 9 min was equivalent. According to the comprehensive results of the mortality rate and the positive mutation rate, the optimal time for UV mutagenesis of the strain HNCMBJ-P-01 was 12 min (Figure 1).

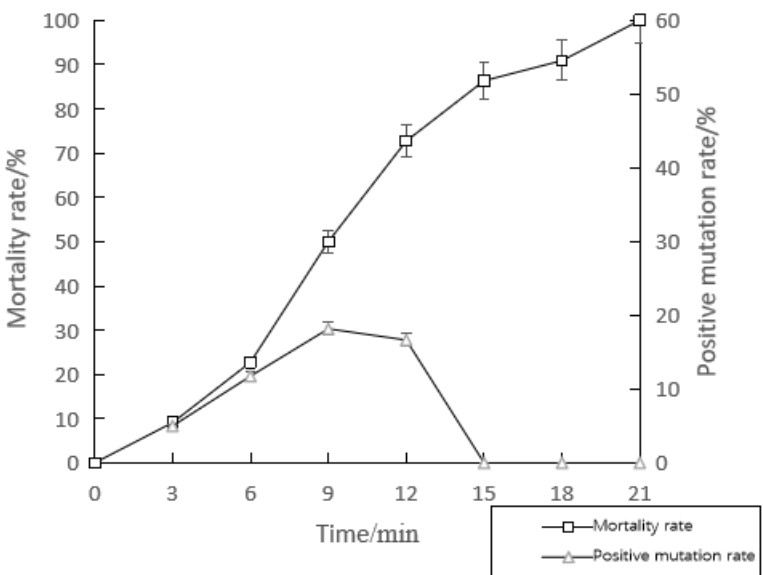

**Figure 1.** The mortality and positive mutation rate of strain HNCMBJ-P-01 by UV mutagenesis. Bars represent standard errors of three replicates.

### 3.2. Determination of Spore Production of UV-Mutated Strains

After the original strain HNCMBJ-P-01 was subjected to UV mutagenesis for 12 min, the spore suspension was diluted, spread and cultured, and a total of 34 *B. bassiana* colonies were obtained. The obtained colonies were re-inoculated on PDA medium for purification and culture. It was obvious that the sporulation yields of the strains obtained by UV mutagenesis were different. Among them, 19 mutant strains showed lower sporulation yield than the original strain, and 15 mutant strains showed higher sporulation yield than the original strain. Particularly, the sporulation yields of five positive mutant strains were 1.5 times higher than that of the original strain, namely HBUV-01, HBUV-06, HBUV-11, HBUV-22, and HBCV-27 (Table 1).

**Table 1.** Sporulation yields of 34 *B. bassiana* strains by UV mutagenesis.

| Strains | Sporulation Yields ($10^7$ Spores/cm$^2$) | Growth Rate (%) | Strains | Sporulation Yields ($10^7$ Spores/cm$^2$) | Growth Rate (%) |
|---|---|---|---|---|---|
| CK | 5.12 | | HBUV-18 | 7.1 | 38.67 |
| HBUV-01 | 7.81 | 52.54 | HBUV-19 | 4.89 | −4.49 |
| HBUV-02 | 2.54 | −50.39 | HBUV-20 | 4.46 | −12.89 |
| HBUV-03 | 4.96 | −3.13 | HBUV-21 | 4.05 | −20.90 |
| HBUV-04 | 0 | 0 | HBUV-22 | 8.27 | 61.52 |
| HBUV-05 | 4.37 | −14.65 | HBUV-23 | 3.27 | −36.13 |
| HBUV-06 | 11.12 | 117.19 | HBUV-24 | 5.01 | −2.15 |
| HBUV-07 | 6.04 | 17.97 | HBUV-25 | 3.87 | −24.41 |
| HBUV-08 | 5.02 | −1.95 | HBUV-26 | 1.39 | −72.85 |
| HBUV-09 | 3.38 | −33.98 | HBUV-27 | 10.26 | 100.39 |
| HBUV-10 | 5.74 | 12.11 | HBUV-28 | 5.17 | 0.98 |
| HBUV-11 | 7.86 | 53.52 | HBUV-29 | 4.64 | −9.38 |
| HBUV-12 | 3.56 | −30.47 | HBUV-30 | 2.28 | −55.47 |
| HBUV-13 | 5.85 | 14.26 | HBUV-31 | 0 | 0 |
| HBUV-14 | 5.33 | 4.10 | HBUV-32 | 0 | 0 |
| HBUV-15 | 5.38 | 5.08 | HBUV-33 | 5.38 | 5.08 |
| HBUV-16 | 4.28 | −16.41 | HBUV-34 | 6.21 | 21.29 |
| HBUV-17 | 5.96 | 16.41 | | | |

### 3.3. Virulence Determination of UV-Positive Mutant Strains

Table 2 showed the corrected mortality of the positive mutant strains obtained by UV mutagenesis of the fourth instar larvae of *P. bilinealis*. Five positive mutant strains obtained through the primary screening demonstrated certain insecticidal effects on the fourth instar larvae of *P. bilinealis*, and the corrected mortality rate increased gradually with the increase in UV treatment time. On the 10th day, the corrected mortality rate of the original strain HNCM-P-1 was 64.81%, and the $LT_{50}$ value was 7.334 d. The corrected mortality rate of the mutant strains HBUV-01 and HBUV-27 was significantly lower than that of the original strain, the corrected mortality rate of the strain HBUV-11 was comparable to that of the original strain, and the corrected mortality rate of the strains HBUV-06 and HBUV-22 was higher than that of the original strain. The corrected mortality and $LT_{50}$ values of strain HBUV-22 were 1.17 and 0.88 times higher than those of the original strain, respectively, with significant differences.

**Table 2.** Corrected mortalities (%) and $LT_{50}$ of *P. bilinealis* treated with positive UV-mutant strains of *B. bassiana*.

| Strains | Days | | | | | | | | $LT_{50}$/d |
|---|---|---|---|---|---|---|---|---|---|
| | 3 | 4 | 5 | 6 | 7 | 8 | 9 | 10 | |
| ck | 0 | 0 | 1.67 | 3.33 | 5 | 6.67 | 8.33 | 10 | |
| HNCM-P-1 | 8.33 | 16.67 | 28.81 | 43.11 | 50.87 | 57.14 | 61.82 | 64.81 [c] | 7.334 [c] |
| HBUV-01 | 5.00 | 11.67 | 15.25 | 25.86 | 36.84 | 46.43 | 54.54 | 57.41 [e] | 8.254 [a] |
| HBUV-06 | 6.67 | 15.00 | 27.11 | 43.11 | 52.63 | 60.71 | 67.27 | 70.37 [b] | 7.146 [d] |
| HBUV-11 | 8.33 | 16.67 | 28.81 | 39.66 | 50.87 | 57.14 | 61.82 | 62.97 [d] | 7.397 [c] |
| HBUV-22 | 10.00 | 23.33 | 33.90 | 50.01 | 61.40 | 69.65 | 74.55 | 75.92 [a] | 6.514 [e] |
| HBUV-27 | 3.33 | 16.67 | 30.51 | 37.93 | 45.61 | 51.78 | 54.54 | 55.56 [f] | 7.849 [b] |

Different superscript letters indicate values significantly different according to Duncan's multiple range tests ($p < 0.05$).

### 3.4. Determination of Microwave Mutagenesis Time

Figure 2 shows the relationship between microwave mutagenesis time and strain mortality and positive mutation rate. The mortalities of the strains increased gradually with the increase in microwave irradiation time. When the microwave irradiation time was 120 s, the mortality rate was 100%. From 0 to 60 s, the positive mutation rate of the

strains increased with the increase in microwave irradiation time. When the microwave mutagenesis time was 60 s, the mortality rate was 61.90%, and the positive mutation rate was the highest at this time, which was 20.83%. According to the comprehensive results of the mortality rate and positive mutation rate, the optimal microwave mutagenesis time was established as 60 s.

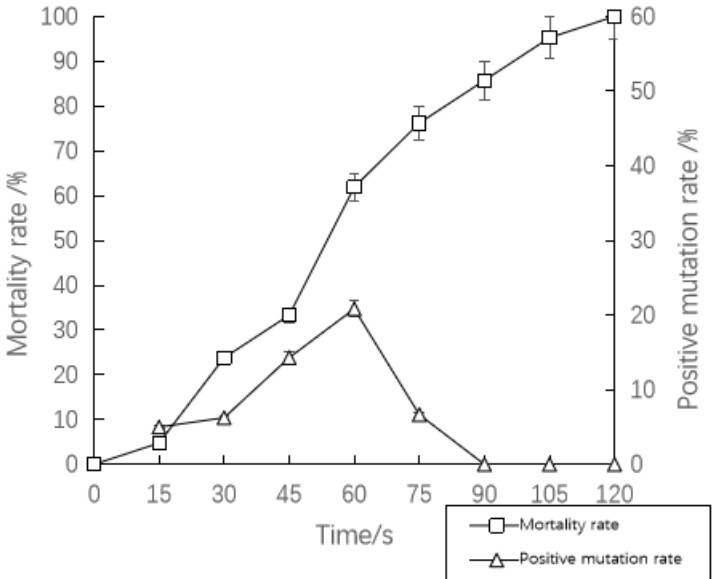

**Figure 2.** The mortality and positive mutation rate of strain HBUV-22 by microwave mutagenesis. Bars represent standard errors of three replicates.

### 3.5. Determination of Spore Production of Microwave-Mutated Strains

After the UV-mutated strain HBUV-22 was subjected to microwave mutagenesis for 60 s, the spore suspension was diluted, spread and cultivated, and a total of 47 *B. bassiana* colonies were obtained. The obtained colonies were re-inoculated on PDA medium for purification and culture. The sporulation yields of strains are shown in Table 3. Among them, 30 mutant strains showed lower sporulation quantity than the original strain, and 17 mutant strains showed higher sporulation quantity than the original strain. Especially, the sporulation yields of six positive mutant strains were 1.5 times higher than that of the original strain, namely HBWB-12, HBWB-27, HBWB-29, HBWB-36, HBWB-40, and HBWB-44.

**Table 3.** Sporulation yields of 47 *B. bassiana* strains by microwave mutagenesis.

| Strains | Sporulation Yields ($10^7$ Spores/cm$^2$) | Growth Rate (%) | Strains | Sporulation Yields ($10^7$ Spores/cm$^2$) | Growth Rate (%) |
|---|---|---|---|---|---|
| HBUV-22 (CK) | 10.55 | | HBWB-24 | 6.67 | −36.78 |
| HBWB-01 | 5.54 | −47.49 | HBWB-25 | 8.21 | −22.18 |
| HBWB-02 | 7.28 | −31.00 | HBWB-26 | 11.79 | 11.75 |
| HBWB-03 | 7.83 | −25.78 | HBWB-27 | 18.47 | 75.07 |
| HBWB-04 | 11.72 | 11.09 | HBWB-28 | 4.23 | −59.91 |
| HBWB-05 | 11.25 | 6.64 | HBWB-29 | 25.66 | 143.22 |
| HBWB-06 | 6.67 | −36.78 | HBWB-30 | 6.84 | −35.17 |
| HBWB-07 | 9.21 | −12.70 | HBWB-31 | 5.15 | −51.18 |
| HBWB-08 | 8.72 | −17.35 | HBWB-32 | 13.96 | 32.32 |
| HBWB-09 | 5.91 | −43.98 | HBWB-33 | 13.51 | 28.06 |
| HBWB-10 | 7.82 | −25.88 | HBWB-34 | 7.23 | −31.47 |
| HBWB-11 | 8.02 | −23.98 | HBWB-35 | 6.82 | −35.36 |
| HBWB-12 | 18.29 | 73.36 | HBWB-36 | 20.08 | 90.33 |
| HBWB-13 | 14.44 | 36.87 | HBWB-37 | 11.64 | 10.33 |

**Table 3.** *Cont.*

| Strains | Sporulation Yields ($10^7$ Spores/cm$^2$) | Growth Rate (%) | Strains | Sporulation Yields ($10^7$ Spores/cm$^2$) | Growth Rate (%) |
|---|---|---|---|---|---|
| HBWB-14 | 8.93 | −15.36 | HBWB-38 | 0 | −100.00 |
| HBWB-15 | 5.98 | −43.32 | HBWB-39 | 4.64 | −56.02 |
| HBWB-16 | 12.52 | 18.67 | HBWB-40 | 16.55 | 56.87 |
| HBWB-17 | 6.76 | −35.92 | HBWB-41 | 3.94 | −62.65 |
| HBWB-18 | 13.82 | 31.00 | HBWB-42 | 4.98 | −52.80 |
| HBWB-19 | 9.91 | −6.07 | HBWB-43 | 12.17 | 15.36 |
| HBWB-20 | 5.62 | −46.73 | HBWB-44 | 21.76 | 106.26 |
| HBWB-21 | 0 | −100.00 | HBWB-45 | 5.23 | −50.43 |
| HBWB-22 | 7.23 | −31.47 | HBWB-46 | 7.19 | −31.85 |
| HBWB-23 | 14.81 | 40.38 | HBWB-47 | 6.46 | −38.77 |

*3.6. Virulence Determination of Microwave Positive Mutant Strains*

Table 4 showed the corrected mortality of the positive mutant strains obtained by microwave mutagenesis of the fourth instar larvae of *P. bilinealis*. It can be seen that the virulence of the six positive mutant strains to the fourth instar larvae of *P. bilinealis* was different, and the strain HBWB-44 exhibited the strongest virulence, e.g., the 10-day corrected mortality rate reached 80.00% and the $LT_{50}$ value was 5.622 days, which was, respectively, 1.23 and 0.76 times higher than those of the original strain HNCM-P-1.

**Table 4.** Corrected mortalities (%) and $LT_{50}$ of *P. bilinealis* treated with positive microwave-mutant strains of *B. bassiana*.

| Strains | Days | | | | | | | | $LT_{50}$/d |
|---|---|---|---|---|---|---|---|---|---|
| | 3 | 4 | 5 | 6 | 7 | 8 | 9 | 10 | |
| CK | 2.5 | 5 | 7.5 | 7.5 | 7.5 | 10 | 12.5 | 12.5 | |
| HBUV-22 | 10.26 | 21.05 | 37.84 | 48.65 | 59.46 | 66.67 | 71.43 | 71.43 [d] | 6.398 [e] |
| HBWB-12 | 5.13 | 13.16 | 27.03 | 43.24 | 54.05 | 61.11 | 68.57 | 71.43 [d] | 6.929 [c] |
| HBWB-27 | 12.82 | 15.79 | 24.32 | 37.84 | 51.35 | 61.11 | 62.86 | 65.71 [e] | 7.107 [b] |
| HBWB-29 | 7.69 | 15.79 | 37.84 | 45.95 | 59.46 | 66.67 | 71.43 | 74.29 [c] | 6.376 [e] |
| HBWB-36 | 7.69 | 18.42 | 32.43 | 45.95 | 56.76 | 63.89 | 74.29 | 77.14 [b] | 6.573 [d] |
| HBWB-40 | 5.13 | 10.53 | 18.92 | 32.43 | 37.84 | 41.67 | 42.86 | 48.57 [f] | 8.485 [a] |
| HBWB-44 | 12.82 | 23.68 | 43.24 | 62.16 | 67.57 | 69.44 | 74.29 | 80.00 [a] | 5.622 [g] |
| CXBJ-01 | 5.13 | 13.16 | 32.43 | 45.95 | 54.05 | 66.67 | 71.43 | 71.43 [d] | 6.845 [c] |
| CXBJ-03 | 7.69 | 18.42 | 37.84 | 48.65 | 59.46 | 66.67 | 74.29 | 77.14 [b] | 6.141 [f] |

Different superscript letters indicate values significantly different according to Duncan's multiple range tests ($p < 0.05$).

*3.7. Stability Determination of UV–Microwave Mutated Strain*

The UV–microwave mutated strain HBWB-44 was subcultured five times, and the sporulation and insecticidal virulence of each generation were determined. As shown in Table 5, there was no significant difference in insect virulence, indicating that the genetic performance of the strain was stable, which is worthy of further research.

**Table 5.** Sporulation yields and corrected mortality of subcultured HBWB-44 strain.

| Generation | Sporulation Yields ($10^7$ Spores/cm$^2$) | Corrected Mortality (%) |
|---|---|---|
| 1 | 21.76 | 80.00 |
| 2 | 20.22 | 77.54 |
| 3 | 23.47 | 78.23 |
| 4 | 21.86 | 80.75 |
| 5 | 20.61 | 78.83 |

There were no significant differences between strains based on one-way analysis of variance in both sporulation yields and corrected mortality.

### 3.8. Strain Heat-Resistance Assay

The screened highly virulent mutant strain HBWB-44, original strain HNCM-P-1, *B. bassiana* CXBJ-01, and *B. bassiana* CXBJ-03 were subjected to 48 °C heat stress and heat-resistance test. It can be seen that the spore germination rate of the four strains was higher under 6 min and 12 min stress, the original strain and the mutant strain had stronger heat resistance, and the spore germination rate remained above 60% under 30 min of heat stress. Strain CXBJ-01 and strain CXBJ-03 showed poor heat tolerance and the conidial germination rates were lower than 10% after 36 min of heat stress (Table 6). The HBWB-44 strain was selected for the forest control effect test.

**Table 6.** Spore germination rate of different strains under 48 °C heat stress.

| Strains | Spore Germination Rate (%) | | | | | | | $LT_{50}$ (min) |
|---|---|---|---|---|---|---|---|---|
| | 6 min | 12 min | 18 min | 24 min | 30 min | 36 min | 42 min | |
| HNCM-P-1 | 95.44 | 91.25 | 84.39 | 75.12 | 64.17 | 51.37 | 45.89 | 37.158 [b] |
| HBWB-44 | 96.65 | 91.54 | 85.92 | 74.26 | 66.18 | 58.46 | 51.97 | 43.541 [a] |
| CXBJ-01 | 92.82 | 74.67 | 53.92 | 36.51 | 14.35 | 5.74 | 0 | 20.844 [d] |
| CXBI-03 | 97.33 | 86.45 | 68.21 | 54.83 | 33.82 | 9.61 | 0 | 25.167 [c] |

Different superscript letters indicate values significantly different according to Duncan's multiple range tests ($p < 0.05$).

### 3.9. Indoor Virulence Assay of B. bassiana Wettable Powder

The indoor measurement results showed that after 10 days of treatment, the insecticidal effects of 100 times *B. bassiana* dilution, *B. bassiana* spore suspension, 1000 times *B. bassiana* dilution (Lvhai), and 1000 times matrine dilution were similar (Figure 3). The insecticidal effect of the matrine dilution was significantly higher than that of other groups in the first 2 days. The insecticidal effect of 100 times *B. bassiana* dilution, *B. bassiana* spore suspension, and 1000 times *B. bassiana* dilution (Lvhai) was mainly shown after 4 days, which might be due to the slow effect of biocides.

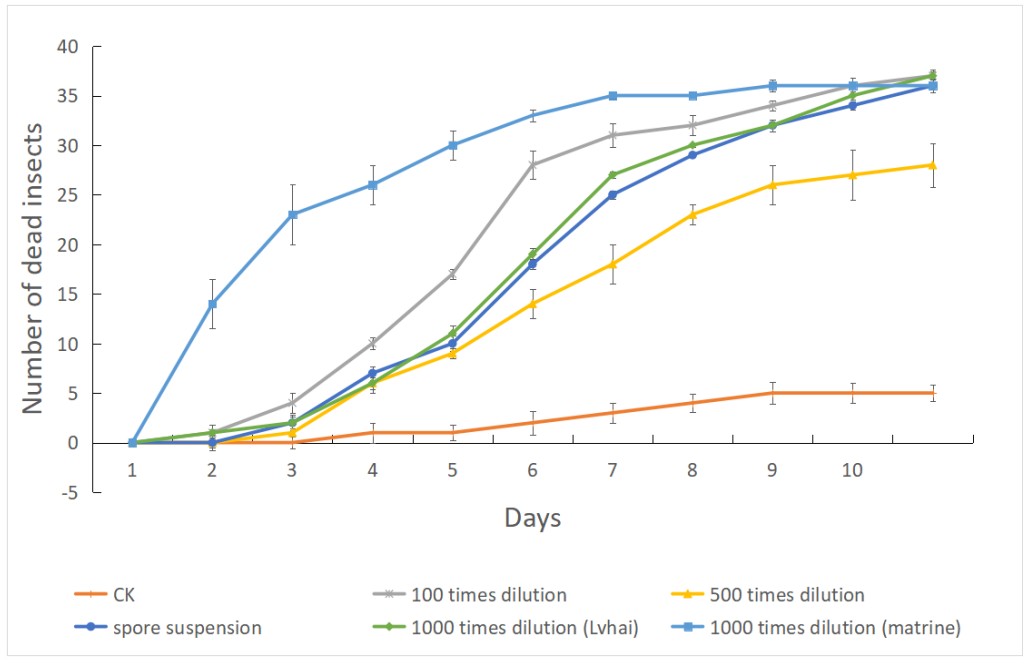

**Figure 3.** Mortality trend of *P. bilinealis* larvae with different treatment solutions. Bars represent standard errors of three replicates.

### 3.10. Forest Control Effect of B. bassiana Wettable Powder

　　As can be seen from Figure 4, after spraying different agents for 2–10 days, the insect population reduction rate of each treatment group showed an increasing trend to different degrees, among which the 1000-fold matrine treatment group and the 100-fold *B. bassiana* treatment group showed the most obvious effect. The population decline rate of the control group showed a negative increasing trend, which was due to the natural reproduction of *P. bilinealis*. As can be seen from Figure 5, the relative control effects of different treatment groups showed an upward trend within 2 to 10 days, and tended to be stable within 8 to 10 days. Therefore, the duration of the treatment agents was about 10 days. For controlling and applying pesticides to the forest pests of *D. odorifera*, multiple spraying should be carried out in a cycle of 10 days. After 10 days of treatment, the highest control effects of 100 times *B. bassiana* WP dilution, 500 times *B. bassiana* WP dilution, *B. bassiana* spore suspension, 1000 times *B. bassiana* WP dilution (Lvhai) and 1000 times matrine dilution were 60.98%, 41.51%, 43.76%, 50.83%, and 74.87%, respectively. Compared with 500 times *B. bassiana* WP dilution, *B. bassiana* spore suspension, and 1000 times *B. bassiana* WP dilution (Lvhai), the insecticidal efficiency of 100 times *B. bassiana* WP dilution was significantly enhanced and significantly weakened compared to that of 1000 times matrine dilution. The control effect of *B. bassiana* HBWB-44 wettable powder on the leaf-eating pests of *D. odorifera* was second only to that of the botanical fungicide matrine.

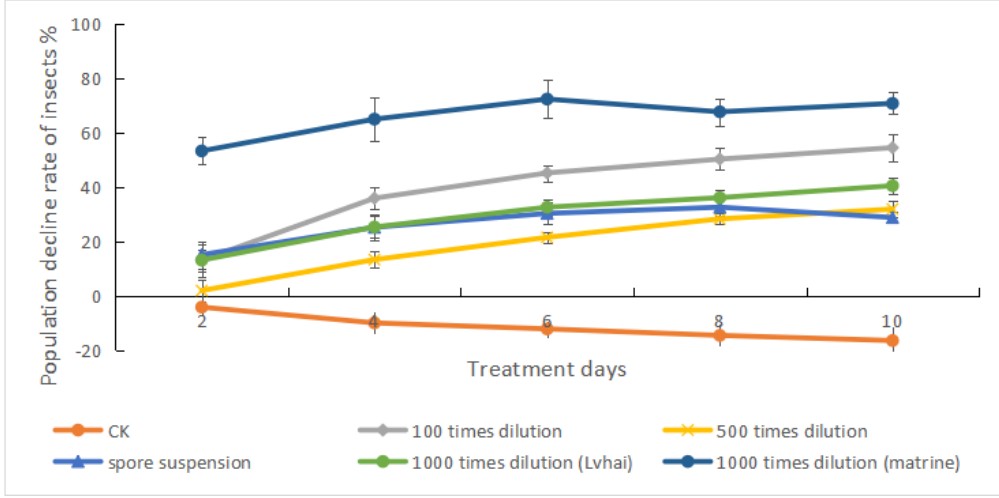

**Figure 4.** Effects of different treatment agents on the population decline rate of *P. bilinealis* larvae. Bars represent standard errors of three replicates.

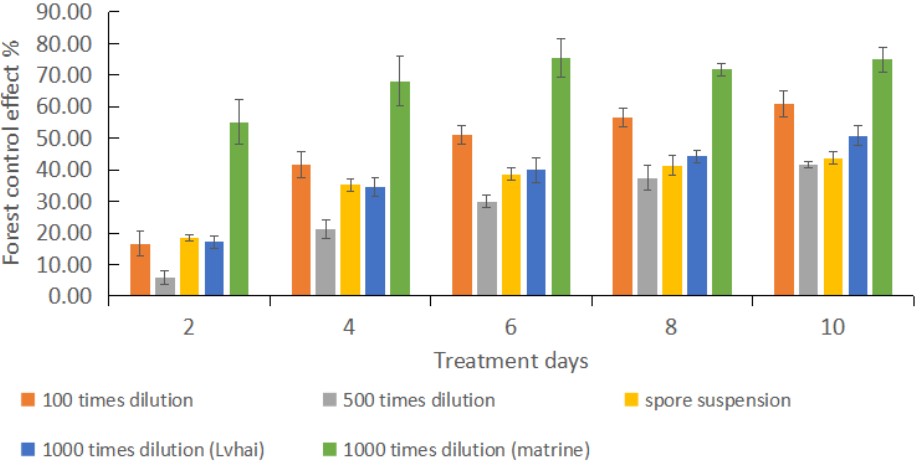

**Figure 5.** Forest control effect of spraying different treatment agents on *P. bilinealis* larvae. Bars represent standard errors of three replicates.

## 4. Discussion

Researchers have already recognized the role of entomogenous fungi in the control of agricultural pests [35,36]. The practical application of entomogenous fungi depends largely on strain screening. The characteristics of a strain with potential for application include high virulence, large spore production, etc., of which high virulence is regarded as the most important point. In order to screen out strains with excellent characteristics, the number of strains to be primarily screened should be large; otherwise, the target strains may not be obtained. Additionally, under the long-term mutagenesis of a single mutagenic agent, the strain often produces "fatigue effect", which reduces the sensitivity of the strain to the mutagenic agent and reduces the mutagenesis effect. However, the use of a variety of mutagenic agents for compound mutagenesis treatment can make the strain obtain the best mutagenesis effect. We carried out UV mutagenesis on the original strain, and screened the positive mutant HBUV-22 through the spore yield and virulence assays. Subsequently, the strain HBUV-22 was mutated by microwave mutagenesis. Similarly, the target strain HBWB-44 was obtained through the spore yield and virulence tests. There have been many reports on the screening of heat-resistant strains. It is simple to use the 48 °C water bath stress method to determine the heat resistance of strains [37]. Li Hongwen's team successfully measured the thermo tolerance of several *B. bassiana* strains using this method [38]. After subculture, the mutant strain HBWB-44 in our study showed good stability in spore production, pathogenicity, and heat resistance, and had good production and application prospects.

*B. Bassiana*, a kind of important insect pathogenic fungi, has been developed into the environment-friendly fungal insecticide and widely used in agriculture and forestry. In order to facilitate transportation and field application, it is usually prepared in specific formulations. *B. Bassiana* formulations are made by processing the effective agents and other components in a certain proportion. Wang Haihong et al. developed the *B. Bassiana* wettable powder with a high control effect on *Frankliniella occidentalis*, and the control effect reached more than 74% [39]. Xiao-Ying Pu et al. used a formulation of *B. Bassiana* conidia and imidacloprid effectively to control the false-eye leafhopper [40]. We explored *B. bassiana* wettable powder in the pest control of *D. odorifera* for the first time. In forest control, the relative control effect of *B. bassiana* WP (Lvhai) was lower than that of *B. bassiana* HBEB-44 WP, which may be due to the poor heat resistance of *B. bassiana* spores, which is not conducive to survival in the high-temperature environment in Hainan forests. The relative control efficiency of spore suspension was obviously lower than that of the 100 times *B. bassiana* WP, which was different from the results of indoor measurements. It may be that the adjuvant in the wettable powder made the spores have better adhesion, and the conidia can adhere to the leaves better, or the UV-protective agent enables the spores to survive in high-temperature environments. This could be further explored in future research.

The problems to be further explored include: (1) Single application of *B. bassiana* WP has similar effects to general biological insecticides, but the infection cycle is long and the effect is slow. It can be considered to enhance the insecticidal effect by combining with green and low-toxic chemical pesticides or botanical agents. (2) The colonization of *B. bassiana* HBWB-44 in pests is not yet clear. How it affects the viability of *P. bilinealis* can be further studied to explore the physiological and biochemical responses of the host.

## 5. Conclusions

The highly virulent strain HBWB-44 was obtained by UV–microwave mutagination of *B. bassiana* HNCMBJ-P-01. The results showed that the strain had stable genetic performance and strong heat resistance. The prepared WP was tested for indoor virulence and forest control effect. Results of the indoor virulence assay showed that the insecticidal effects of 100 times *B. bassiana* dilution, *B. bassiana* spore suspension, 1000 times *B. bassiana* dilution (Lvhai), and 1000 times matrine dilution were similar after treatment for 10 days. The forest control effect test was carried out in a Hainan *D. odorifera* plantation base. Results showed that the control effect of *B. bassiana* wettable powder 100 times solution reached 60.89%,

second only to the botanical fungicide matrine. The *B. bassiana* WP had a good control effect on *D. odorifera* leaf-eating pests, and can be used in the artificial forest to prevent and control the occurrence of *D. odorifera* insects.

**Author Contributions:** Conceptualization, X.N. and G.Z.; methodology, C.L. and H.L.; formal analysis, C.W. and C.L.; writing—original draft preparation, X.N. and C.L.; writing—review and editing, J.L. and Y.X.; supervision, G.Z.; project administration, G.Z. and Y.L.; funding acquisition, J.L. All authors have read and agreed to the published version of the manuscript.

**Funding:** This research was funded by the National Key Research and Development Program Project (2018YFD0600202), National Promotion Project of Forestry and Grassland Scientific and Technological Achievements (No. 2020133124), and the Postgraduate Science and Technology Innovation Fund of Central South Forestry University (No. 2022CX02066 and No. CX202102007). Funders had no role in the study design, data collection and analysis, publication decisions, or manuscript preparation.

**Institutional Review Board Statement:** Not applicable.

**Informed Consent Statement:** Not applicable.

**Data Availability Statement:** All relevant data are within the paper.

**Conflicts of Interest:** The authors declare no conflict of interest. The funders had no role in the design of the study; in the collection, analyses, or interpretation of data; in the writing of the manuscript; or in the decision to publish the results.

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
