# Peer review of "Breeding of Highly Virulent Beauveria bassiana Strains for Biological Control of the Leaf-Eating Pests of Dalbergia odorifera"

_forests, doi:10.3390/f14020316_

Round 1

Reviewer 1 Report

Comments and Suggestions for Authors

Manuscript ID: forests-2130048

The paper entitled “reeding of Beauveria bassiana Strains Highly Virulent to the Leaf-eating Pests of Dalbergia odorifera and Development of New Insecticides” was carefully reviewed.

This manuscript is fairly well written, and needs extensive revision. Here are some critical points:

1.      The relevance of the research has not been clearly identified and discussed;

2.      Technical clarifications: many deficiencies in Materials & Methods section compromise the understanding of ideas and concepts and affect the quality of the paper;

3.      The statistical analyses are not presented;

4.      Poorly prepared discussion lacking appropriate interpretation. The Discussion should be concise and relevant to the results. Furthermore, some statements are simply incorrect and/or are not supported by any scientific literature.

5.      Many "key" references have been missed and the list of references is not up to date;

6.      Extensive editing of English language and style are required.

Detailed comments:

Title

-          Breeding of Beauveria bassiana Strains Highly Virulent to the Leaf-eating Pests of Dalbergia odorifera and Development of New Insecticides”. This title should be revised and improved. Please delete “and Development of New Insecticides” as it is not within the scope of the study.

Abstract

-          Line 13: Indicate the common name of Dalbergia odorifera.

-          Line 14: “in my country's national standard”. Please correct.

-          Avoid repeating the full scientific names of species in the text. Use the abbreviation (e.g. replace Dalbergia odorifera by D. odorifera, and Beuveria bassiana by B. bassiana).

-          Line 20: “with high spore yield and high toxicity were”. You mean high virulence instead of toxicity? Please correct the sentence.

-          Lines 25-27: Add quantitative data.

Introduction

-          Line 37: “5genera and 8categories”. Please correct.

-          Line 38: Avoid repeating the full scientific names of species in the text. Use the abbreviation (e.g. replace Dalbergia odorifera by D. odorifera).

-          Line 51: “Spodoptera triceratops, Spodoptera ash, Spodoptera nigra”. Indicate the full scientific names of insect pests.

-          Line 49: “the damage of leaf-eating pests is getting more and more serious”. This statement requires references. Also indicate the percentage of damage for each of the 3 pests listed (Spodoptera triceratops, Spodoptera ash, and Spodoptera nigra).

-          Line 56: “as a common class of microbial pesticides” Not relevant. Please delete.

-          Lines 56-58: Add a reference for this statement.

-          Lines 59-63: Delete this paragraph as it does not contain relevant information.

-          Lines 66-67: “Strains isolated from nature often cannot meet the requirements of industrial production”. Add references for this statement.

-          The introduction section still needs to be deeply improved. Enlarge the state of the art by adding other relevant and recent works in the field.

-          The novelty of the work is not clear in the introduction. Complete the introduction with a paragraph or two to explain the novelty of the work and indicate the scope and purpose of this study.

Materials and methods

-          Line 73: Delete “The” from the sub-title.

-          Line 74: “P. bilinealis”. This is the first mention of this pest in the text! The authors used this insect to perform all the experiments but did not introduce or show its economic importance, distribution, damage, etc., in the Introduction Section. So please rewrite a paragraph or two to present relevant information about P. bilinealis.

-          Lines 79-80: “This strain was isolated from the zombies”? You mean cadavers? Please correct. The cadaver of which insect? please indicate these details.

-          Lines 87-93: 2.2. Experimental design: “Ultraviolet mutagenesis, microwave mutagenesis and compound mutagenesis were ……….obtained by screening, and screened out the mutant strains with better virulence effect on P. bilinealis”. This paragraph is not well written. Please rewrite it in a good manner to avoid any misunderstandings.

-          Lines 94-95: "In order to improve the sporulation rate and insecticidal effect of B. bassiana WP, different carriers and wetting agents were used as main additives in this experiment”. You should list these additives with the concentrations used to prepare B. bassiana formulations, and explain why you considered each additive.

-          Line 96: “the wettable powder was applied to the Dalbergia odorifera plantation”. Indicate the rates for each application.

-          Line 113: “2.4. UV mutagenesis and screening of high virulent strains”: I have some concerns:

o   B. bassiana strains were UV irradiated and then treated with microwave. What were the optimal ultraviolet mutagens conditions (as stated in line 113)?

o   What were frequency and power of the microwave oven?

o   After treatment, the strains were incubated for how long and at what temperature? Did you consider a specific pH value to create selective pressure?

-          Line 126: “2.5. Microwave mutagenesis and screening of high virulent strains”. Did you test the efficacy of the mutant strains on P. bilinealis after microwave mutagenesis? If so, explain how.

-          Line 167: indicate the coordinates of the experimental sites.  

-          Line 169: “and five points sampling method was adopted”. Please explain in details the sampling technique.  

-          Line 171: Why not measure the damage associated with P. bilinealis after application of B. bassian strains?

-          I have major reservations about the data analysis. The authors did not present the statistical analyses.

Results:

-          Lines 174-176: “B. bassiana strain HNCMBJ-P-01 was subjected to UV mutagenesis to investigate the relationship between UV irradiate time and the mortality rate and the positive mutation rate”. Delete this sentence because it was already listed in the "Materials and Methods" section.

-          Line 196: “Table 1. Sporulation yields of 34 strains by UV mutagenesis”. 34 strains of Beauveria bassiana? Please correct.

-          Line 212: “Table 2. Corrected mortalities and LT50 of Plecoptera bilinealis treated with positive mutant strains”. strains of Beauveria bassiana? Please correct.

-          Line 212: What do you mean by “a, b, c & d” in table 2?

-          Line 237: “Table 3. Sporulation yields of strains by microwave mutagenesis”. strains of Beauveria bassiana? Please correct.

-          Line 246: “Table 4. Corrected mortalities and LT50 of Plecoptera bilinealis treated with positive mutant strains”. The title of table 4 is similar to that of table 2 “Corrected mortalities and LT50 of Plecoptera bilinealis treated with positive mutant strains”. Please rectify.

-          Line 247: What do you mean by “a, b, c, d, e…” in table 4?

-          Line 264: Replace LT50/min by LT50 (min) in Table 6.

-          Line 264: What do you mean by “a, b, c, d” in table 6?

-          Line 294: “Forest control sffect %” (Axe Y of the Figure 5). Please correct.

-          Lines 286-289: “The highest control 286 effects of 100 times B. bassiana WP dilution, 500 times B. bassiana WP dilution, B. bassiana spore suspension, 1000 times B. bassiana WP dilution (Lvhai) and 1000 times matrine dilution were 60.98%, 41.51%, 43.76%, 50.83% and 74.87%, respectively”. Is there any statistically significant difference between the treatments?

-          The quality of all the figures should be improved.

Discussion:

-          The Discussion Section here looks like as concluding remarks (one paragraph)!

-          No reference was cited in the Discussion Section!

-          The authors did not compare their results in depth with others reported in the literature. I recommend rewriting the discussion section to more accurately understand and express the relevance of the findings in regard to what was previously known.

Reviewer 2 Report

Check notes into the manuscript and specially with the references. Attached the pdf document with notes.

Round 2

Reviewer 1 Report

The authors have done extensive research on the topic and have presented interesting work in the revised version of the manuscript. They have responded to all comments and the article can be considered for publication.

Author Response

Thank you so much for helping us improve our work!